# An Integrated Approach for the Thermal Maturity Modeling Re-Assessment of an Exploration Well in the Hellenides Fold and Thrust Belt

**Vagia Ioanna Makri** [1,2,*], **Spyridon Bellas** [1,*], **Georgia Moschou** [2] **and Nikos Pasadakis** [1,2]

1   Institute of GeoEnergy, Foundation for Research and Technology-Hellas (FORTH/IG), 73100 Chania, Greece
2   School of Mineral Resources Engineering, Technical University of Crete, 73100 Chania, Greece
*   Correspondence: vayanna@ipr.forth.gr (V.I.M.); spyrosbellas@ipr.forth.gr (S.B.)

**Abstract:** Utilizing geological and geochemical data, we re-assessed the thermal maturity of the Lower Cretaceous Vigla shales of the AY-3 well, located in the Internal Ionian geotectonic zone of Greece, using 1D thermal maturity modeling. Vigla shales primarily containing kerogen type I to II, incorporated within alternations of carbonates, cherts, and marly limestones, were selected as the main source rock intervals. Biomarkers and Rock-Eval data were used on top of vitrinite reflectance data for the calibration of the 1D model. Hopane and sterane isomerization ratios for the Vigla shales appear to hold values of 58–64% and 44–49%, respectively, while vitrinite reflectance ranges from 0.61% to 0.71% and Tmax between 431 and 451 °C. One-dimensional thermal maturity modeling suggests that lower Cretaceous Vigla shales entered the oil window in early Miocene times and reached the expulsion onset during the middle Miocene. Additionally, thermal modeling estimates the overburden eroded thickness to range between 2.1 and 2.6 km. This unravels the pre-eroded shape of this part of the belt of Miocene times alongside the burial history of the area and its evident relation to the hydrocarbon potential. This assessment comprises a step towards the understanding of the belt and the different timings of hydrocarbon generation in the External Hellenides.

**Keywords:** maturity modeling; Western Greece; biomarkers; Ionian zone; Fold and Thrust Belt

## 1. Introduction

The wider study area is one previously awarded for Exploration & Production (E & P) block, called "Arta-Preveza", and it is located in Western Greece, one of the frontier areas of the eastern Mediterranean region. It was included in the last licensing round of onshore exploration activities of 2014 [1], and it was also part of the first International Licensing Round back in 1996 [2]. This area lies within the Ionian geotectonic zone (Figure 1) of the Hellenic Fold and Thrust Belt (HFTB) of westward prograding intracontinental thrusting [3–6], and extends northwards to Albania, where exploration and production have been active for decades.

Several geochemical studies have been carried out on oil seeps, shows, and source rocks of the HFTB dating back to 1984, i.e., [7–12]. Rigakis and Karakitsios [8,9] have employed thermal maturity modeling in several well locations to understand the effect of the HFTB tectonics in hydrocarbon generation and expulsion. They suggested that the oil window for the Agios Georgios-3 (AY-3) well, located in the Arta syncline, within the Internal Ionian zone, is reached at 3.45 km measured depth (MD), while for the Dragopsa-1 well located in the Central Ionian zone the data suggest immaturity across the penetrated interval. This was later developed in [13], which proved the co-existence of thin- and thick-skinned tectonics, and [14] which suggested a 40% shortening of the belt at its northern Greek part, which descends to 20% to the south, in Peloponnese. The latter suggested that active kitchens in the Ionian zone are in the underthrusts, beneath the penetrated intervals. According to a study by Marin et al. [15], the overburden thickness erosion was estimated

to be 2.5 km in Dragopsa-1 well, revealing the pre-erosional structure of this location. Later, Refs. [12,16] added to this picture, combining the tectonic regime with the source rock maturity, showing that outcrop maturities across Western Greece range from immature to mature, while well penetrated intervals reach higher maturities across the HFTB.

The AY-3 well initially modeled by [9] suggested good calibration with the Vigla shale layers A and B (among the two main source rocks of the Ionian basin) entering the oil window during Early Miocene, considering an overall erosion of 1.76 km. Recently, the model of the AY-3 well was re-assessed [17] with similar outcomes to [9], with an overall overburden erosion of almost 2 km and a Lower Oligocene hydrocarbon generation.

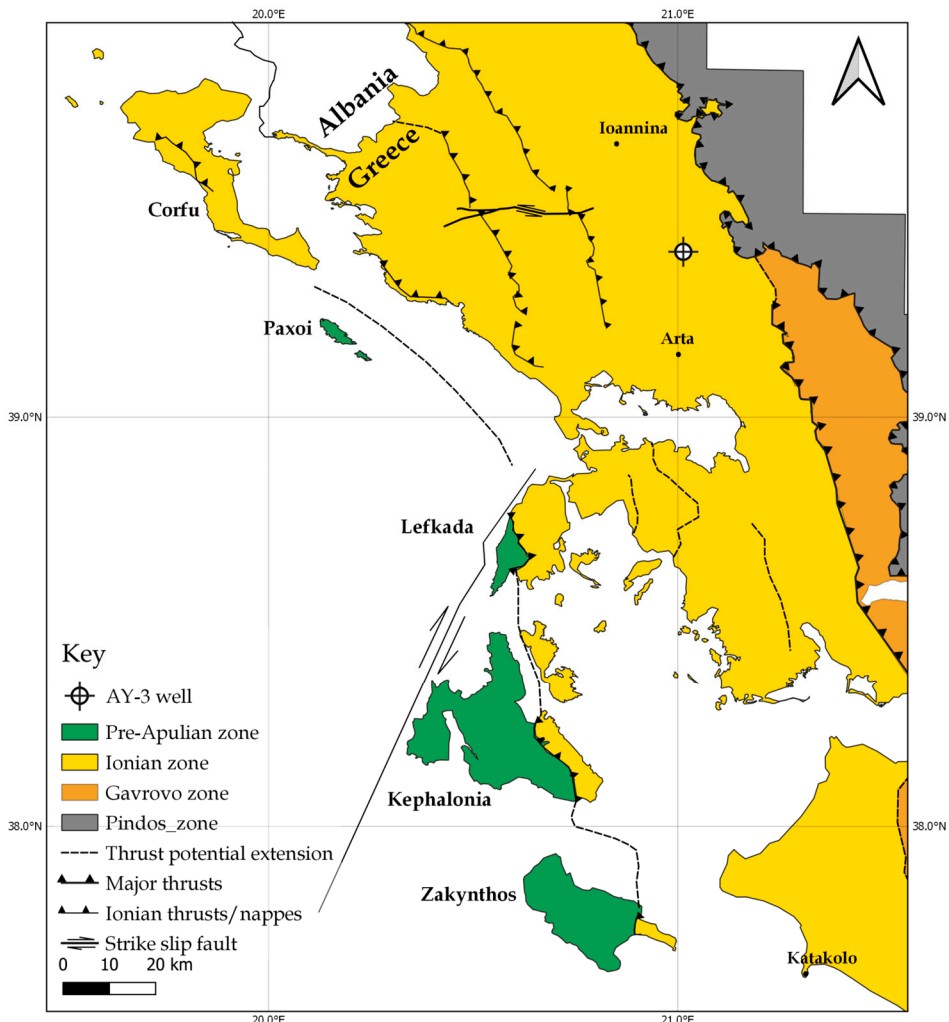

**Figure 1.** Simplified map of the geotectonic zones of the HFTB in Western Greece, illustrating the well under assessment (AY-3). Modified after [5,14,16,18].

This study aims to improve the understanding of the pre-erosional structure of the AY-3 area, which has directly affected the source rock maturity and hydrocarbon generation timing. For this purpose, a W-E transect was studied, cross cutting the AY-3 well. One-dimensional thermal modeling of the well was utilized to reconstruct the overburden eroded thickness. Biomarker isomerization ratios and vitrinite reflectance (VR: Ro% and VRE$_{Tmax}$ calculated from Rock-Eval pyrolysis Tmax (°C)), from the literature data [9] and from samples analyzed for this study, were used as thermal indicators for the calibration of the model.

## 2. Geotectonic Setting

### 2.1. Structural Setting

The HFTB dominates Western Greece in a NNW-SSE trend [5,19]. It was generated by the subduction of the NeoTethyan oceanic crust, which separated the African and Eurasian Plates during the Late Cretaceous [20]. This resulted in the formation of the Alpine orogenic belt, part of which represents the Dinarides–Albanides–Hellenides FTB [4,21,22] with the collision continuing throughout most of the Tertiary as a westward prograding deformation [6,14,20]. The compressional regime resulted in the deformation of the external Hellenides, the uplift of the entire Hellenides orogenic belt, and the development of a foreland basin at the eastern Apulian margin [23–25]. Subduction is currently taking place in the southern part of this orogenic system, while collision occurs in the Kephalonia right-lateral strike-slip fault [26].

Based on subcrustal earthquakes, this is a highly seismic zone which was recently interpreted as a pro-to-step fault that evolved together with the Epirus Pliocene fragmentation [27,28]. According to fault plane solutions [29], its strike-slip direction changes slightly to the east (NW of Lefkada island), in agreement with the counterclockwise Adria microplate rotation.

Studies [30–32] have shown that the Hellenides were affected by a clockwise rotation after a Lower Miocene phase of N-S extension. This was later followed by a Middle Miocene compressional regime and the subsequent collision of Western Greece with the Apulian platform, giving rise to the middle Ionian thrust [30]. These have resulted in a complex tectonic regime, with several seismically active right lateral strike-slip faults [33], adding to the seismicity of the area.

Several models have been proposed, i.e., Refs. [5,13,14,18,24,34,35], regarding the structural setting of the HFTB. The most prominent pertain to a combination of thin and thick-skinned tectonics in the area with major decollement horizons at the base level of the Triassic evaporites, which are distinctly evident at the thrust boundary between the Pre-Apulian and Ionian zones [5,14,18,20,36].

Due to this compressional event, the HFTB was divided into NNW-SSE lying geotectonic zones [8,21], separated by major thrusts. In this study, we will focus on the Ionian geotectonic zone, which is bordered in the east by the Pindos and Gavrovo thrusts, which were activated during Eocene–Early Miocene and Oligocene–Early Miocene, respectively, and in the west by the Ionian thrust, which was activated during the Pliocene–Quaternary (Figure 1) [5,8,14,20,35,37–39].

### 2.2. Geological Setting

Since the pioneering work of [40,41], many geological studies have been made relevant to the Ionian geotectonic zone in Western Greece, i.e., [3]. Their main focus was the Mesozoic and Cenozoic evolution of the Epirus area ([18], see references therein). Later, [37] combined marine seismic data to approach the offshore development of this zone to the west. Many authors [9,19,42,43] have provided a detailed overview of the Mesozoic tectonostratigraphy and organic geochemistry of the Ionian geotectonic zone.

The well under consideration is located in Western Greece, and more specifically in the Epirus region, where it penetrates Mesozoic and Cenozoic sediments of the Ionian geotectonic zone. The external part of the HFTB can be divided into three thrust-bound tectonostratigraphic zones which extend northwards into Albania. From east to west, these are the Gavrovo, Ionian and pre-Apulian (or Paxi) zones, i.e., [3,8,21,23]. The Ionian zone in Greece is subdivided into three partly thrust-bound belts or sub-zones (Internal, Central/Middle, and External (Figure 1)) [3,12,18,23,44].

The tectonostratigraphic evolution of the Ionian zone is reflected on the deposition of, principally, four distinct geological sequences, each one indicative of a different tectonic regime [19,25]:

(1) A pre-rift sequence represented by thick (>1.5 km) Early Jurassic platform "Pantokrator" Limestones. The latter overlies the also thick (>2 km) Early to Mid-Triassic evaporites through the "Foustapidima" Limestones of Ladinian–Rhetian age [45].

(2) A syn-rift sequence (Pliensbachian–Tithonian) deposited during extensional faulting and halokinesis of the Triassic evaporites, which caused the formation of the Ionian basin and its internal syn-rift differentiation into smaller sub-basins characterized by asymmetric half-graben geometry and various carbonates thickness accumulation [19,46]. Toarcian–Tithonian syn-rift pelagic deposits in the half-grabens are correlated to global oceanic anoxic events (T-OAE) [47].

(3) A post-rift sequence (Early Cretaceous–Eocene) deposited after the cessation of extensional faulting (Early Berriasian break-up). It is marked by an unconformity at the base of the "Vigla" Limestones [25]. It consists of deep-marine carbonate facies [48] intercalated with bedded chert and shales. Again, the organic rich layers of this formation are correlated to global OAE (Paquier OAE) [47].

The Mesozoic carbonate dominant succession [49] passes upwards via transitional beds [50] to the Flysch synorogenic sedimentation (mostly siliciclastic turbidites), which began at the Eocene–Oligocene boundary [18,44,45]. Until the Early Miocene, the basin was filled by submarine fan deposits, in response to the compressional event [44,51], while periodically the basin was sourced from the western margin of the Ionian basin as well [52]. The Ionian zone's overview of hydrocarbon systems was refined by [53], while overall several contributions have been made to its better understanding, i.e., [23,53–55]. Source and reservoir rocks of the Ionian zone have been mainly documented by the following authors, [1,5,8,19,49,56]. Five are the main source rock intervals:

- Aptian–Turonian (Cretaceous) Vigla shales;
- Callovian–Tithonian Upper Posidonia beds;
- Toarcian–Aalenian (Jurassic) Lower Posidonia beds;
- Time equivalent marls at the base of the Ammonitico Rosso (Toarcian);
- Shallow-water organic-rich shales within the Triassic evaporites.

In addition, two more source rock intervals have been reported recently, the first one by [57] of Late Oligocene–Early Miocene, and the second by [12] of Late Triassic–Lower Jurassic age.

The area of interest focuses on the transect shown in Figure 2. It is trespassing the AY-3 well, which penetrates the subsurface up to 4288 m, while the measured, from the ground level, depth (MD) is almost equal to the true vertical depth (TVD) [9]. In the west it is bordered by a backthrust, while in the east the whole stratigraphic section of the Ionian zone from Triassic to Oligocene is evident, yet with only the Oligocene flysch cropping out [18,58,59]. Herein, since detailed geochronological information for the well is not available, Vigla shales are referred to as Lower Cretaceous and Posidonia shales as Toarcian–Upper Jurassic. Two Vigla shale layers have been reported in the well, Vigla A and Vigla B. The former corresponds to the shallower shale layer, and the latter to the deeper (Figure 2b).

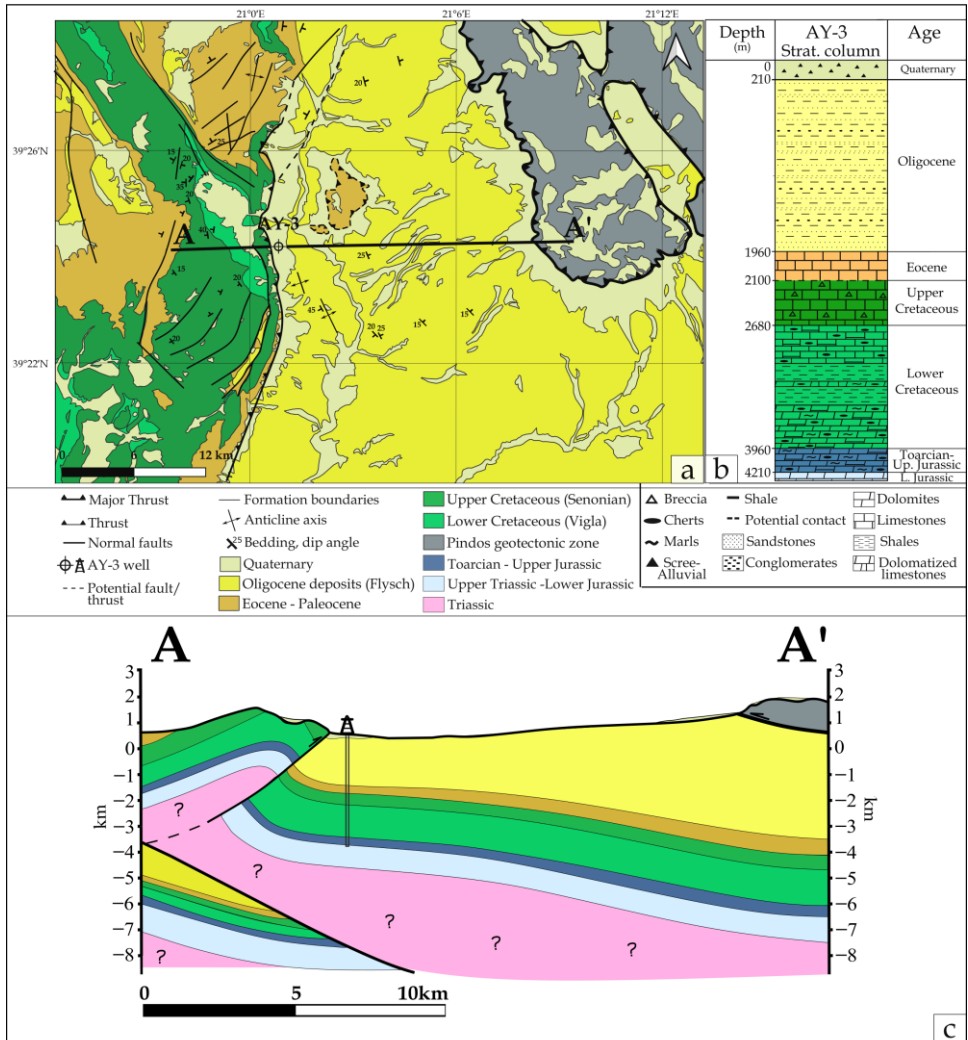

**Figure 2.** (**a**) Geological map of the AY-3 well location area in the Internal Ionian geotectonic zone (compilation of [15,16,18,58,59]). (**b**) Stratigraphic column of the AY-3 well, reconstructed after [9] and (**c**) Simplified cross-section (A-A') in W–E direction crossing a backthrust sequence, the AY-3 well and the Pindos thrust from west to east. Question marks "?" indicate uncertainty about the related displayed layers.

### 3. Materials and Methods

*3.1. Samples*

Own samples were analyzed for this study (G1–G7), by the methods discussed in this chapter. These are core samples of the Lower Cretaceous Vigla shales and the Toarcian–Upper Jurassic Posidonia shales from the AY-3 well, comprising potential source rocks.

On top of that, available geological and geochemical sample data were retrieved from [9] to support the 1D thermal maturity model of the AY-3 well. These are samples G8–G28 and are presented in Section 4.

Regarding the literature data, the Rock-Eval parameters were determined using a Rock-Eval II (RE2) pyrolizer by the former Public Petroleum Corporation-Exploration and Production of Hydrocarbons (DEP-EKY S.A.) (present Helleniq Energy S.A.) and biomarkers were determined by Gas Chromatography-Mass Spectrometry (GC-MS) in the United States Geological Survey (USGS) using a Hewlett-Packard 5710A GC [9]. Lastly, the %Ro was determined by optical microscopy at the Coal Research Center, Canada, using a Zeiss MPM II microscope, while at least 50 vitrinite particles were identified for reflectance measurements when present [9].

*3.2. Methods*

3.2.1. Rock-Eval 6 Pyrolysis (RE6)

Rock-Eval pyrolysis was performed using a Rock-Eval VI$^{®}$ (RE6) anhydrous open system pyrolizer (Vinci Technologies; [60]) following the basic setup for the organic matter analysis [61]. Representative quantities of finely ground (<250 μm) and cooked samples went through sequential pyrolysis and oxidation [62]. The resulting volatile hydrocarbons were detected and quantified using a flame ionization detector (FID). The pyrolysis trend was isothermal, starting from 300 °C and increasing with a 25 °C/min ramp to reach a peak of 650 °C. Following pyrolysis, the oxidation was carried out from 300 to 850 °C with a heating rate of 20 °C/min. As a result, S1, S2 (mgHC/grock) (HC stands for hydrocarbons), S3 (mgCO$_2$/grock), Tmax (°C), TOC (%wt), HI (mgHC/gTOC) and OI (mgCO$_2$/gTOC) were determined [60,62]. These are the free hydrocarbons in the sample, the hydrocarbons formed by the thermal cracking of kerogen, the CO$_2$ yield during the thermal cracking of kerogen, the temperature at which the maximum amount of hydrocarbons is generated, the total organic carbon content, the amount of hydrogen relative to the amount of organic carbon present in the sample and the amount of oxygen relative to the amount of organic carbon present in the sample, respectively [62].

3.2.2. Gas Chromatography–Mass Spectrometry (GC-MS)

Grounded samples were extracted for 24 h by a Soxhlet apparatus using dichloromethane:methanol (DCM:MeOH, 90:10 v.). The extract was de-asphalted using n-pentane (nC$_5$), and the isolated maltenes were separated into saturated, aromatic, and polar (NSO) fractions using open-column chromatography. This separation was carried out in serological pipets (5 mL) filled with a mixture (5:2 w.) of SiO$_2$ (100–200 mesh, 30 Å, Davisil type 923) and Al$_2$O$_3$ (70–230 mesh). Saturates were eluted with nC$_5$ (7 mL), aromatics with toluene (6.5 mL) and NSO compounds with a 60:40 *v/v* mixture of toluene–methanol. GC-MS analysis was carried out on the saturated fractions on an Agilent 7890A Gas Chromatograph coupled to an Agilent 5975E Mass Spectrometer with an automatic liquid sampler.

The GC was equipped with an Agilent capillary column HP-5MS UI (60 m × 250 μm × 0.25 μm). The saturated fractions were dissolved in n-hexane (0.5 mL), and 15 μL of internal standard of n-Dodecane-d26, Fluoronaphthalene, 3-Fluorophenanthrene, n-Hexadecane-d34, 2-Fluorochrysene and 5β(H) Cholane (Chiron mixture S-4121-ASS-IO) were added. The oven temperature was programmed at 40 °C for 2 min, followed by a 20 °C/min ramp to 200 °C, a second 2 °C/min ramp to 300 °C and a final isothermal time of 30 min. The transfer line was set at 280 °C, and the MS source was set at 230 °C.

Sterane and terpane biomarkers were identified and analyzed based on selected ion and mass spectra monitoring. The ions used were the *m/z* 191 for terpanes and *m/z* 217–218 for steranes, with *m/z* 191 and *m/z* 217 used for their calculation (Figure 3) [63]. Specific biomarkers related to organic matter maturity were identified and isomerization ratios were calculated based on them. These are the C$_{29}$ ααα 20S/(20S + 20R) sterane, the C31 and C32 22S/(22S + 22R) homohopane and the Ts/(Ts + Tm) (Table 1). In this study, the C32 22S/(22S + 22R) homohopane and the C29 20S/(20S + 20R) sterane were used as calibration parameters to the 1D thermal maturity modeling.

3.2.3. Thermal Maturity Modeling

Thermal history and burial of the AY-3 well, Internal Ionian geotectonic zone, was performed using the PetroMod 1D 2020.1, Schlumberger software package (Schlumberger; version 2020.1). The stratigraphy used is shown in Figure 2b. Relevant lithologies were built with the PetroMod Lithology editor, based on cutting and core lithological information provided by [9]. The assumptions used for the modeling are: (1) layer thicknesses, lithology, and ages are derived from [9]; (2) an almost instantaneous thrusting duration [64]; (3) a sediment–water interface temperature (SWIT) of 15.84 °C based on [65]; (4) paleo-water depth and sea-level changes were neglected, as overburden thickness is the chief factor in thermal evolution [66]; (5) the thermal maturity model was constrained by organic thermal

indicators (VR and biomarkers both from [9] and own samples analyzed in this study, with VR being calibrated using the Burnham Easy%RoDL [67,68]). Lastly, (6) present-day heat flow (HF) values are around 40 mW/m$^2$ based on [69,70], and (7) the HF trends are generated based on [71] considering a Jurassic rifting (Pliensbachian–Tithonian) that reaches an HF of 66 mW/m$^2$ in the Upper Jurassic.

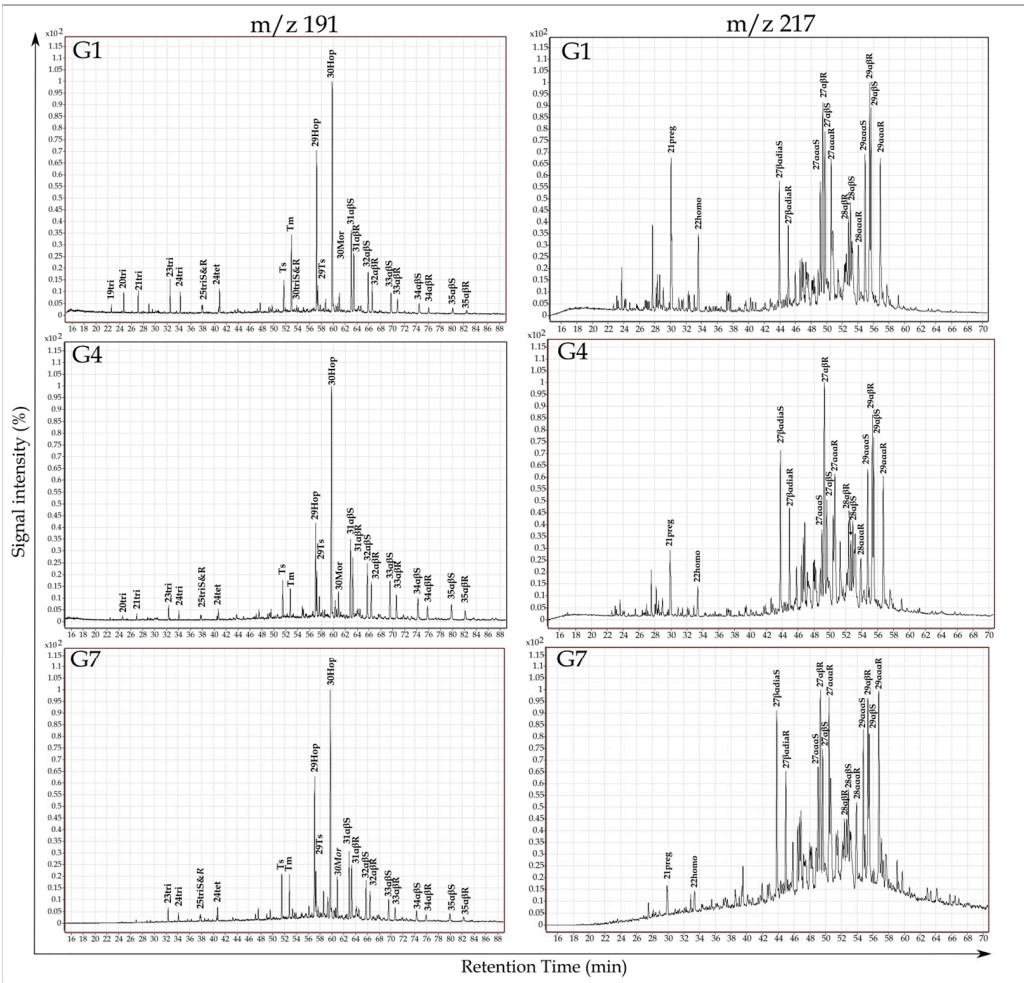

**Figure 3.** GC-MS chromatograms of the ions $m/z$ 217 and $m/z$ 191 for the samples G1, G4 and G7. For the $m/z$ 217, the biomarkers are as follows: 21Preg, C21-5α(H) sterane; 22homo, C22 sterane; 27βαdiaS & R, C27βα20(S) & (R) diasterane; 27αααS, C27ααα20S sterane; 27αβR, C27αββ20R; 27αβS, C27αββ20S sterane; 27αααR, C27ααα20R sterane. The same pattern also holds for the C28 and C29 steranes. For the $m/z$ 191: 23–24tri, C23-C24-Tricyclic terpane; 25triS & R, C25-Tricyclic terpane (S) & (R); 24tet, C24 17,21-Secohopane (C24); Ts, 18α(H)-22,29,30 Trisnorhopane; Tm, 17α(H)-22,29,30 Trisnorhopane; 29Hop, 17α(H),21β(H)-30-Norhopane; 29Ts,18α(H)-30-Norneohopane; 30Hop, 17α(H),21β(H)-Hopane; 30Mor, 17β(H),21α(H)-Moretane; 31–35αβS and 31–35αβR, 22S-17α(H),21β(H)-Bishomohopane-22S-17α(H),21β(H)-Pentakishomohopane and 22R-17α(H),21β(H)-Bishomohopane-22R-17α(H), 21β(H)-Pentakishomohopane.

## 4. Results

### 4.1. Source Rock Quality and Potential

#### 4.1.1. Rock-Eval 6 Pyrolysis

It is evident from Table 1 that HI values range from 347 to 544 mgHC/gTOC for the Lower Cretaceous Vigla shales, and OI values range between 5 and 54 mgCO$_2$/gTOC as illustrated in a van Krevelen diagram in Figure 4. Vigla shales are considered as the primary source rock in the area, and within the AY-3 well their TOC values range from

0.57 to 11.35%wt while the ratio S2/S3 ranges from 8.5 to 81 and the S2 from 2.19 to 58.16 mgHC/grock. As evident in Figure 5, Vigla shales A and B are clearly differentiated from the rest formations across depth. These suggest a kerogen type I-II according to [72] (Figure 4). This pattern supports Rigakis [9], who analyzed and characterized a large amount of cutting and core data across the depth of this well, proving a kerogen type I-II. The analysis of the Oligocene flysch layers obtains values between 55 and 283 mgHC/gTOC and 11 and 81 mgCO$_2$/gTOC for HI and OI, respectively. Along with the S2/S3, which ranges between 1.5 and 13.6 and the S2 between 0.16 and 1.49 mgHC/grock, a kerogen of mainly type III is proven. Lastly, Tmax values for the Vigla shale layers vary between 431 and 451 °C, suggesting oil window maturity [72].

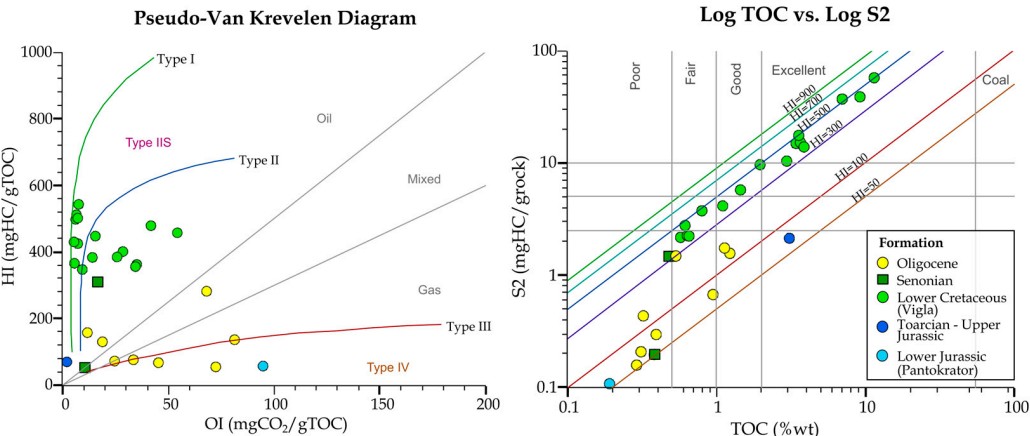

**Figure 4.** (**Left**): Pseudo van Krevelen diagram showing hydrogen (HI; 100xS2/TOC in mgHC/gTOC) and oxygen indices (OI; 100xS3/TOC in mgCO$_2$/gTOC). Kerogen type lines represent typical kerogen maturity paths based on [73]. (**Right**): Plot of the TOC (%wt) versus S2 (mgHC/grock). HI annotated in lines is measured in mgHC/gTOC [72,74]. Legend applies for both plots.

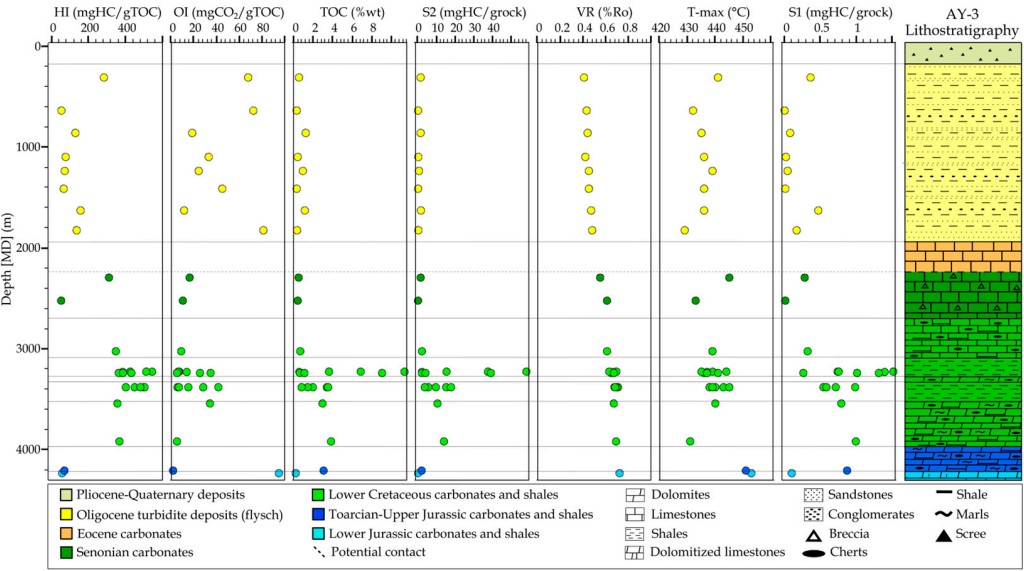

**Figure 5.** Multi-depth plot illustrating several geochemical parameters; Tmax, S1, S2, HI, OI, TOC, with measured depth, related to the simplified stratigraphic column of the well AY-3. The units of the included parameters are shown on the figure and are discussed in Section 3.2.1 The different colors incorporated are related to the stratigraphy of the well, illustrated on the right-hand side.

Increased VR and variance in RE values are evident in the topmost part of the Oligocene flysch compared to its lowermost part (Figure 5). Additionally, a high OI

corresponds to the Lower Jurassic Pantokrator sample (Figure 4), which proves to be inert according to RE parameters (very low HI, S1, S2 and TOC) (Figures 4 and 5).

To reinforce the one-dimensional maturity model, VR was approached according to [75] as a function of Tmax (°C) from RE6 in the samples analyzed for this study (G1 to G6). As low-mature or immature samples pose difficulties when calculating their VR, Wust et al. [75] provided an equation with a good fit compared to the %Ro trend published by [9], and therefore the $VRE_{Tmax}$ corresponds to the equivalent vitrinite reflectance values derived from Tmax (Table 1).

### 4.1.2. Gas Chromatography–Mass Spectrometry

Seifert and Moldowan [76] have discussed the use of biomarkers as thermal maturity indicators, on the basis that during diagenesis biological isomers such as 20R sterane configuration convert into 20S. Isomerization ratios are widely used as thermal maturity indicators. Due to potential problems that biomarkers pose in interpretation, the combination of different ratios and methods is suggested for their rational use [77].

According to the values obtained for the Lower Cretaceous Vigla shales (Table 1), the samples lay mainly within the early oil window (Figure 6). This reinforces RE and VR data. All biomarker ratios range from 31 to 65% (Table 1).

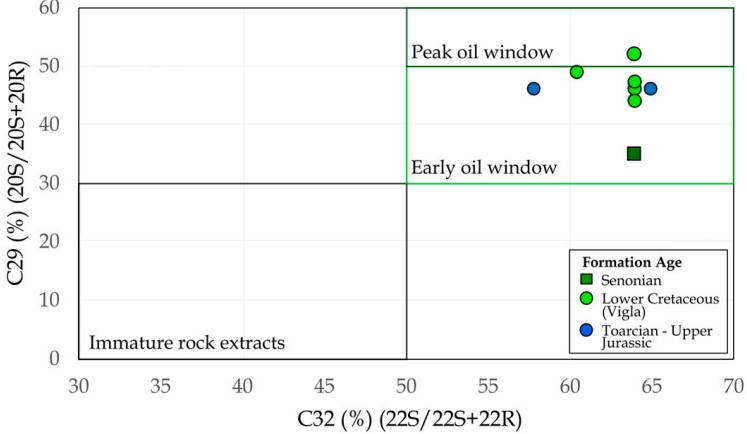

**Figure 6.** Thermal maturity evaluation plot based on biomarker maturity parameters C29 (%) 20S/(20S + 20R) sterane versus C32 (%) 22S/(22S + 22R) hopane as modified after [77].

The least mature samples have low C29 $\alpha\alpha\alpha$ 20S/(20S + 20R) and may lie below the petroleum generation threshold of 40% [78]. This is only evident in the Senonian sample, with the deeper formations surpassing this ratio (Table 1). Other researchers have measured C29 $\alpha\alpha\alpha$ 20S/(20S + 20R) as low as 23% in low maturity oil samples, suggesting that the 40% threshold for this parameter should not be considered de facto for hydrocarbon generation timing across different basins and source rocks [77].

Although easily identified, co-elution is sometimes an issue in terms of C29 sterane identification, and thus homohopane isomerization ratios are preferred for maturity estimations as they are considered to be thermally stable [77]. In general, C31-C35 17$\alpha$-hopane isomerization ratios are used for such characterization, with the use of C29 steranes not being optimal unless they are calibrated for each source rock or basin [77].

Vigla shales have C29% 20S/(20S + 20R) of 44–49%, C31% 22S/(22S + 22R) of 59–64%, C32% 22S/(22S + 22R) of 58–64% and Ts/(Ts + Tm) of 30–57%, with Posidonia shales also ranging within this spectrum. Homohopanes suggest that the oil generation has been reached [77]. On the other hand, Ts/(Ts + Tm) appears to be low for some Vigla shale samples, which might be a source-related issue [79].

**Table 1.** Table including own samples analyzed on this study (G1–G7) and samples from [9] (G8–G28). RE6 pyrolysis parameters for G1–G7 and RE2 for G8–G28. Equivalent VR values VRE$_{Tmax}$ (%) for G1–G7, based on [75]. Vitrinite reflectance values (%Ro) for G8–G28 according to [9]. RE parameters are as follows: S1, S2 (mgHC/grock), S3 (mgCO$_2$/grock), Tmax (°C), HI (mgHC/gTOC), OI (mgCO$_2$/gTOC). Biomarker isomerization ratios are as follows: C29 (%): C29$\alpha\alpha\alpha$20S/(20S + 20R) sterane, C31 and C32 (%): C31 and C32 22S/(22S + 22R), and Ts/(Ts + Tm): 18$\alpha$-22,29,30-trisnoreohopane/(18$\alpha$-22,29,30-trisnoreohopane + 17$\alpha$-22,29,30-trisnorhopane). Cret. refers to Cretaceous and depth to measured depth.

| Sample | Depth (m) | S1 | S2 | S3 | Tmax | HI | OI | TOC | VRE$_{Tmax}$ | %Ro | Ages | Unit | C29 (%) | C32 (%) | C31 (%) | Ts/(Ts + Tm) |
|---|---|---|---|---|---|---|---|---|---|---|---|---|---|---|---|---|
| G1 | 3225 | 0.74 | 37.45 | 0.52 | 439 | 544 | 8 | 6.89 | 0.69 | - | Lower Cret. | Vigla | 49 | 61 | 59 | 30 |
| G2 | 3225 | 1.39 | 2.19 | 0.08 | 437 | 384 | 14 | 0.57 | 0.66 | - | Lower Cret. | Vigla | - | - | - | - |
| G3 | 3236 | 0.26 | 2.29 | 0.22 | 436 | 363 | 35 | 0.63 | 0.65 | - | Lower Cret. | Vigla | - | - | - | - |
| G4 | 3377 | 0.71 | 9.72 | 0.12 | 438 | 498 | 6 | 1.95 | 0.68 | - | Lower Cret. | Vigla | 49 | 61 | 59 | 57 |
| G5 | 3378 | 0.57 | 15.16 | 0.52 | 440 | 449 | 15 | 3.38 | 0.71 | - | Lower Cret. | Vigla | - | - | - | - |
| G6 | 3379 | 0.54 | 5.79 | 0.41 | 439 | 402 | 28 | 1.44 | 0.69 | - | Lower Cret. | Vigla | - | - | - | - |
| G7 | 4068 | - | - | - | - | - | - | - | - | - | Jurassic | Posidonia | 46 | 58 | 57 | 49 |
| G8 | 304 | 0.36 | 1.5 | 0.36 | 441 | 283 | 68 | 0.53 | - | 0.41 | Oligocene | Flysch | - | - | - | - |
| G9 | 632 | 0 | 0.16 | 0.21 | 432 | 55 | 72 | 0.29 | - | 0.43 | Oligocene | Flysch | - | - | - | - |
| G10 | 856 | 0.08 | 1.59 | 0.23 | 435 | 130 | 18 | 1.22 | - | 0.44 | Oligocene | Flysch | - | - | - | - |
| G11 | 1092 | 0.02 | 0.3 | 0.13 | 436 | 76 | 33 | 0.39 | - | 0.42 | Oligocene | Flysch | - | - | - | - |
| G12 | 1232 | 0.04 | 0.68 | 0.23 | 439 | 72 | 24 | 0.94 | - | 0.45 | Oligocene | Flysch | - | - | - | - |
| G13 | 1408 | 0.01 | 0.21 | 0.14 | 436 | 67 | 45 | 0.31 | - | 0.45 | Oligocene | Flysch | - | - | - | - |
| G14 | 1624 | 0.47 | 1.77 | 0.13 | 436 | 158 | 11 | 1.12 | - | 0.47 | Oligocene | Flysch | - | - | - | - |
| G15 | 1820 | 0.17 | 0.44 | 0.26 | 429 | 137 | 81 | 0.32 | - | 0.48 | Oligocene | Flysch | - | - | - | - |
| G16 | 2290 | 0.28 | 1.49 | 0.08 | 445 | 310 | 16 | 0.48 | - | 0.55 | Senonian | Senonian | 35 | 64 | 62 | 32 |
| G17 | 2520 | 0.01 | 0.2 | 0.04 | 433 | 52 | 10 | 0.38 | - | 0.61 | Senonian | Senonian | - | - | - | - |
| G18 | 3020 | 0.32 | 2.26 | 0.06 | 439 | 347 | 9 | 0.65 | - | 0.61 | Lower Cret. | Vigla | - | - | - | - |
| G19 | 3224 | 0.75 | 15.42 | 0.26 | 435 | 425 | 7 | 3.62 | - | - | Lower Cret. | Vigla | 46 | 64 | 63 | 32 |
| G20 | 3225 | 1.51 | 58.16 | 0.74 | 444 | 512 | 6 | 11.35 | - | 0.63 | Lower Cret. | Vigla | - | - | - | - |
| G21 | 3235 | 1.01 | 4.21 | 0.28 | 437 | 386 | 25 | 1.09 | - | - | Lower Cret. | Vigla | 44 | 64 | 64 | 31 |
| G22 | 3236 | 1.31 | 39.1 | 0.49 | 441 | 431 | 5 | 9.07 | - | 0.67 | Lower Cret. | Vigla | 47 | 64 | 63 | 46 |
| G23 | 3378 | 0.98 | 17.74 | 0.25 | 443 | 502 | 7 | 3.53 | - | - | Lower Cret. | Vigla | 52 | 64 | 64 | 65 |
| G24 | 3379 | 0.58 | 3.79 | 0.33 | 445 | 479 | 41 | 0.79 | - | 0.68 | Lower Cret. | Vigla | - | - | - | - |
| G25 | 3540 | 0.79 | 10.49 | 1.01 | 440 | 356 | 34 | 2.94 | - | 0.67 | Lower Cret. | Vigla | - | - | - | - |
| G26 | 3916 | 0.99 | 14.03 | 0.21 | 431 | 366 | 5 | 3.83 | - | 0.69 | Lower Cret. | Vigla | - | - | - | - |
| G27 | 4204 | 0.87 | 2.16 | 0.06 | 451 | 70 | 1 | 3.06 | - | - | Jurassic | Posidonia | 46 | 65 | 62 | 51 |
| G28 | 4229 | 0.1 | 0.11 | 0.18 | 453 | 57 | 94 | 0.19 | - | 0.72 | Lower Jurassic | Pantokrator | - | - | - | - |

### 4.1.3. Thermal Maturity Modeling

Biomarkers identified on ions *m/z* 217 and 191 were used for the calculation of biomarker ratios (Table 1). These ratios were used on top of the VR and RE for the calibration of the AY-3 1D model. Our model can be easily adjusted to the thermal maturity data by varying the present-day HF or the eroded overall thickness. Based on [69,70,80], the approximate HF of the AY-3 wider area is 40 mW/m$^2$. For the model, sensitivity analysis regarding different values of present-day HF and total erosion thickness took place, with the former ranging between 38 and 42 mW/m$^2$ (Figure 7). According to the calibration fitting curves, a present-day HF of 38–40 mW/m$^2$ seems more likely. These scenarios suggest an erosion thickness ranging between 2.1 and 2.6 km for a present-day HF of 40 to 38 mW/m$^2$, respectively.

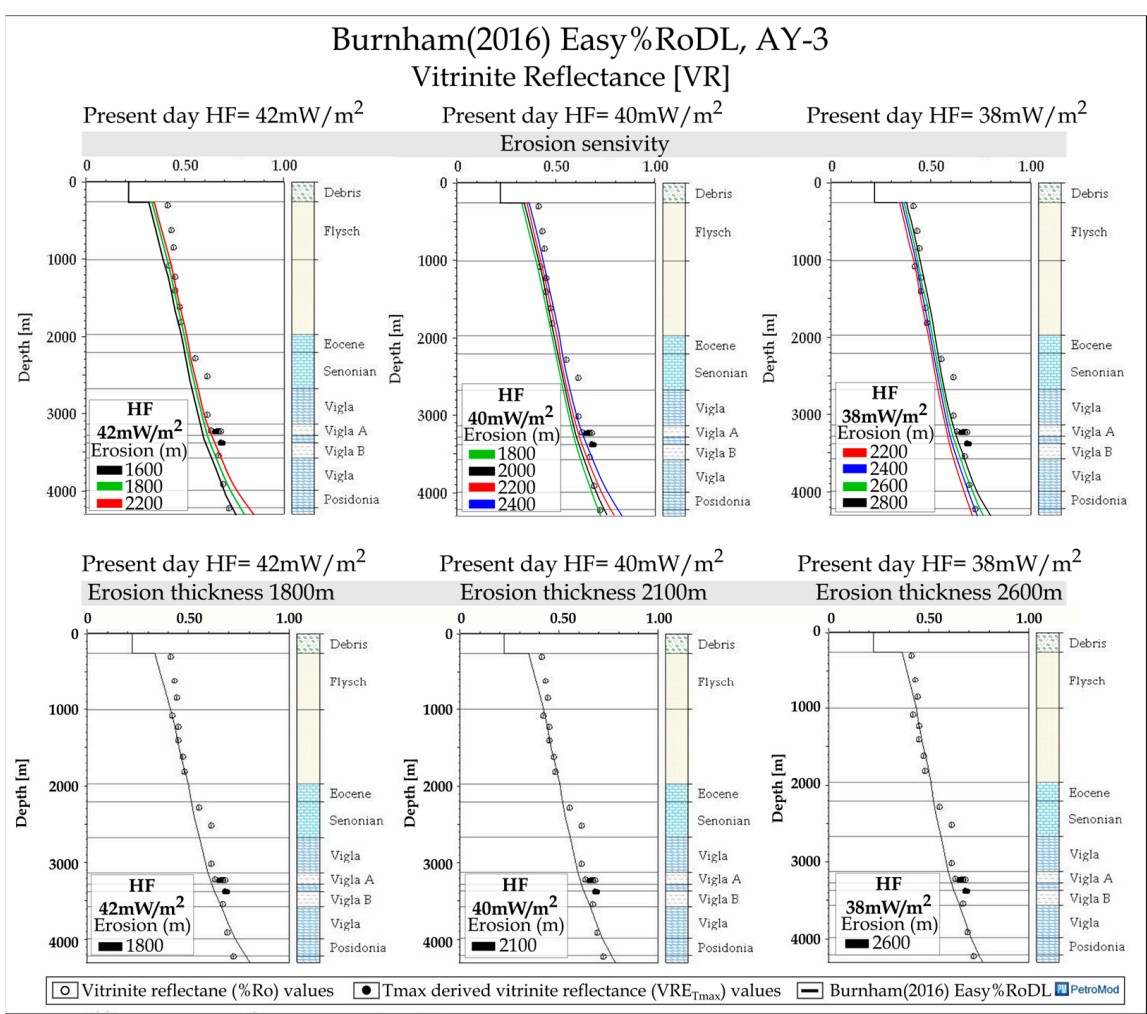

**Figure 7.** Sensitivity analysis results of the AY-3 model. (**Top**): Changing eroded thickness with constant present-day heat flow (HF) scenarios (42, 40, 38 mW/m$^2$). (**Bottom**): Optimum calibration of the VR for the latter HF values. Depth corresponds to ground level MD [9].

For reasons of clarity, the HF scenario selected for the final calibration of the model is the one with 39 mW/m$^2$ present-day HF and 2.4 km overburden erosion. This final fine calibration of the 1D model (Figure 8) suggests a good fit with the VR and biomarkers, using the Pepper and Corvi model for kerogen type I for the Vigla shales [81]. It should be noted that a kerogen type II model [81] was also assessed, providing overall similar outcomes, yet with an earlier generation onset of the order of 2 Ma. It is evident that the biomarker isomerization ratios have a little deviation from the calibration curve, which is introduced in MacKenzie and McKenzie [78] (Figure 8). This is well explained by the reached maturity equilibrium, which ranges between 57 and 62% for the hopane isomerization ratios [77,82], and thus the curve cannot exceed this trend fit. The same holds for sterane isomerization ratios, where the thermal maturity equilibrium is reached at 52–55% [77].

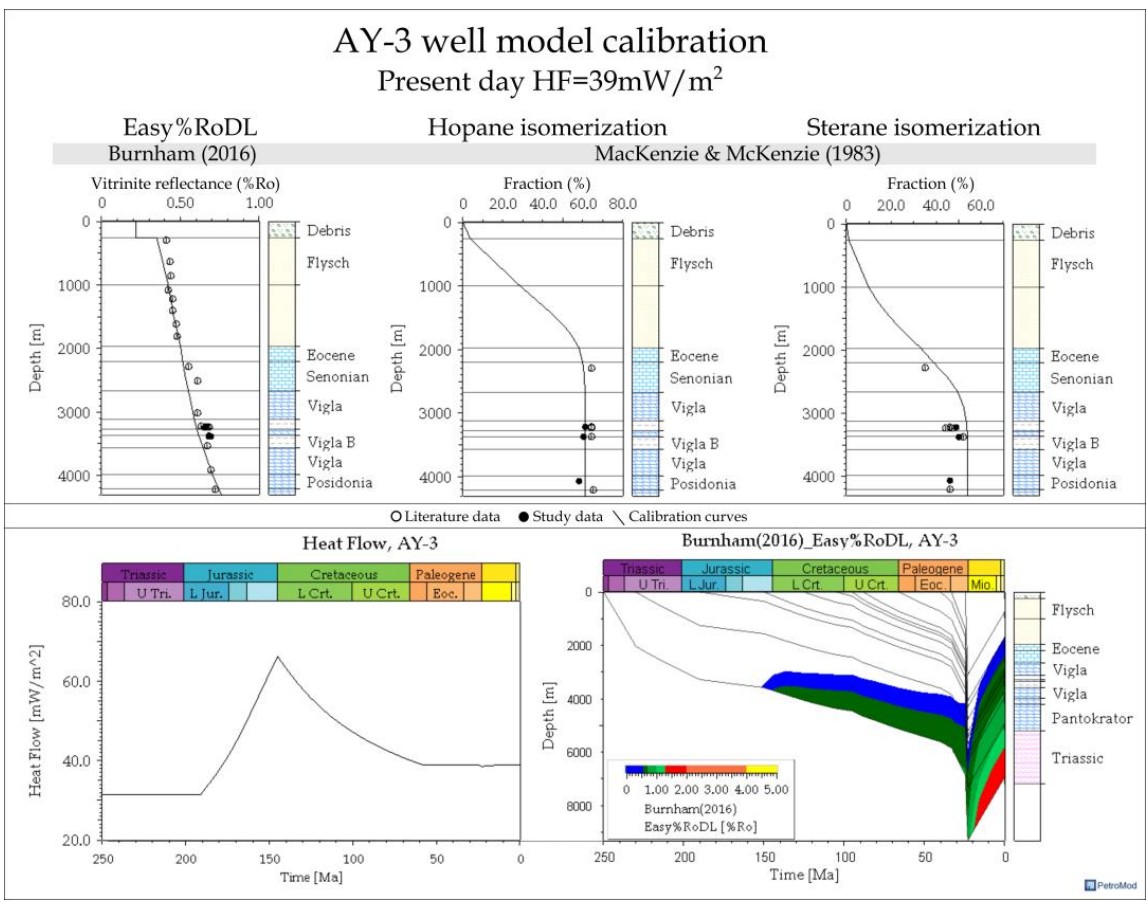

**Figure 8.** Burial and time plots of the AY-3 well. Top row, from left to right: (**left**): Vitrinite reflectance calibration curve using the Easy %RoDL model of Burnham [67]; (**middle**): Hopane isomerization ratio (C32 (%) 22S/(22S + 22R)) calibration curve with activation energy E = 21.5 kcal/mol, frequency factor A = 0e$^{25}$/Ma and hopane isomerization ratio equal to 1.56, based on MacKenzie & McKenzie [78], and (**right**): Sterane isomerization ratio (C29 (%) 20S/(20S + 20R)) curve with E = 21.5 kcal/mol, A = 0e$^{25}$/Ma and hopane isomerization ratio equal to 1.17, based on [78]. Bottom row left: HF time trend based on [71] utilizing a present-day HF of 39 mW/m$^2$ and (**right**): burial history plot of the AY-3 incorporating the Easy %RoDL model [67] and a kerogen type I for the Vigla shales, following the Pepper and Corvi model [81].

## 5. Discussion

The use of multiple thermal maturity parameters is of high importance for the solid maturity understanding of a source rock, as maturity and hydrocarbon generation can be organofacies-, biodegradation- and basin-dependent [77,83,84]. Due to potential uncertainties caused by solely relying on VR data, we proceeded to RE6 and GC-MS analysis (Table 1). The results led to the calculation of VRE$_{Tmax}$ on six more samples [75], and the calculation of biomarker isomerization ratios. The agreement between these and the literature data [9] prove their adequacy for thermal maturity evaluation. As present-day HF is inexplicit yet holding values around 40 mW/m$^2$ [69], sensitivity analysis for the HF and the eroded overburden thickness was employed to the model. This analysis provided feasible present-day HF values and overburden thickness scenarios (Figure 7).

In the Ionian zone, sedimentation began in Triassic with the deposition of evaporites, which later affected the basin evolution [14,18]. During Mesozoic synrift and postrift, mainly deep-water carbonate sedimentation took place, finishing with clastics (flysch). The latter were mainly provided from the east during Oligocene, when the Pindos and Gavrovo thrusts were active. Deformation within the flysch deposits could also be viable should one look at the increased VR values at the top part of the Oligocene flysch, illustrating a higher

maturity trend (Figure 8, top left). During the Alpine orogeny thrust sheets separated the Ionian zone, with all three major thrust sheets accommodating the same Triassic–Oligocene sequence [14,16]. Subsequent backthrusting in the Ionian zone is evident across the belt, as well as on the west of the AY-3 well penetration [16,24].

One-dimensional thermal maturity modeling and sensitivity analysis has suggested an overburden erosion for the AY-3 well of around 2.4 km, which illustrates the burial history shown in Figure 8, implying a maturity increase due to deep burial and subsequent uplift and erosion at post Oligocene times. Cropping out on the flysch (Figure 2), Quaternary clastics are evident and refer to scree and alluvial sediments, suggesting that the geologically youngest formation cropping out is the Oligocene flysch. This justifies the theory that the Ionian thrust itself was active during Pliocene [30], causing the uplift of the belt. The eroded overburden load was defined by the one-dimensional thermal maturity modeling and was incorporated to the transect shown in Figure 9, representing the shape of the area during pre-erosional Early Miocene times. We suggest that the currently eroded overburden thickness corresponds to Oligocene clastic (flysch) deposits, overburden to the present-day stratigraphy, deposited prior to the backthrusting on the west (Figure 9).

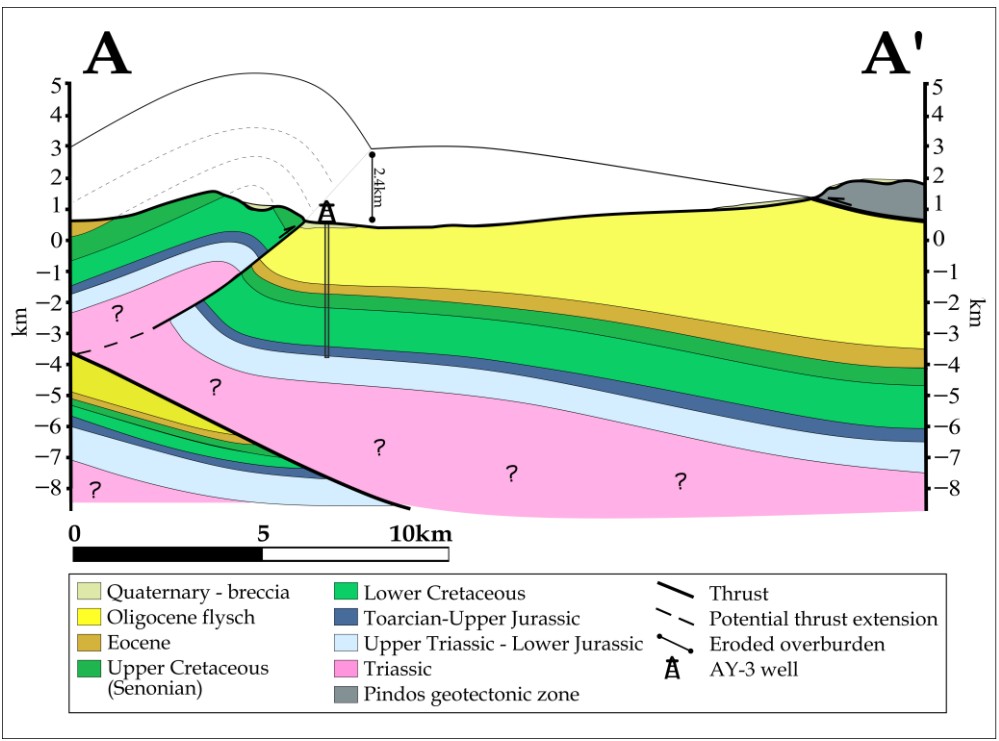

**Figure 9.** Simplified cross section showing the interpreted paleo-shape of the AY-3 area, of the Internal Ionian thrust, in pre-erosion Miocene times. The mean pre-erosion overburden thickness of the AY-3 well location is 2.4 km and it is estimated by thermal maturity modeling.

The HFTB is believed to have undergone a shortening of 40% in the north, decreasing to 20% towards the south [14], while the shortening ceased during the Pliocene [30]. Thermal maturity modeling allows the understanding of the burial history, uplift, and erosion relevant to this shortening. These provide a step towards the reconstruction of the pre-erosional belt shape, and thus the comprehension of the different hydrocarbon generation timings across the HFTB.

## 6. Conclusions

Thermal maturity modeling was employed for the AY-3 well, which penetrates the Internal Ionian geotectonic zone, Epirus, NW Greece, using VR, Rock-Eval and Biomarker data as modeling calibration parameters. Analytical results support that the Lower Cre-

taceous Vigla shales comprise a prominent source rock in the Internal Ionian zone, lying within the early oil window at the well location, with VR ranging from 0.61 to 0.71%Ro and hopane and sterane isomerization ratios between 49 and 65%. Sensitivity analysis was performed on the model with regard to the present-day heat flow and potential overburden erosion, shedding light on the pre-erosional (Miocene) shape of this specific part of the belt and the reached burial depth. The estimated eroded thickness ranges between 2.1 and 2.6 km, with 2.4 km being the most probable case scenario, corresponding to a present-day heat flow of 39 mW/m$^2$. Such a depth was reached prior to Miocene when the dominant compressional and rotational regime resulted in further thrusting and uplift of the belt, which yielded its final shape. Thus, the present-day early maturity stage of the Vigla shales in the AY-3 well location is a consequence of an eroded overburden thickness of approximately 2.4 km.

**Author Contributions:** Conceptualization, V.I.M.; methodology, V.I.M., S.B. and G.M.; software, V.I.M.; validation, S.B. and N.P.; formal analysis, S.B.; investigation, V.I.M. and S.B.; resources, N.P.; data curation, N.P.; writing—original draft preparation, V.I.M.; writing—review and editing, V.I.M. and S.B.; visualization, V.I.M.; supervision, S.B. and N.P.; project administration, N.P.; funding acquisition, N.P. All authors have read and agreed to the published version of the manuscript.

**Funding:** This study was partially funded by Helleniq Energy S.A. within a sponsorship to the Institute of GeoEnergy—Foundation for Research and Technology—Hellas (FORTH/IG), funding number: 7/2020.

**Data Availability Statement:** The literature data utilized (G8–G28) are available in a publicly accessible repository. The literature data presented in this study are openly available in [Didaktorika.gr] at [10.12681/eadd/12079]. The geochemical data of samples G1–G7 of this study are presented in Table 1.

**Acknowledgments:** Special thanks should go to Helleniq Energy S.A. for providing well samples. Andrea Schito is also acknowledged for introducing the complex concept of Fold and Thrust Belts understanding through maturity modeling.

**Conflicts of Interest:** The authors declare no conflict of interest.

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
