# Peer review of "An Integrated Approach for the Thermal Maturity Modeling Re-Assessment of an Exploration Well in the Hellenides Fold and Thrust Belt"

_geosciences, doi:10.3390/geosciences13030076_

Round 1

Reviewer 1 Report

The objective of this paper is to provide an analysis for the pre-erosional structure of the AY-3 area which has directly affected the source rock maturity and hydrocarbon generation timing.

This is an interesting and innovative paper, which provides significant geological information for the western part of Greece. The paper structure is satisfactory, as all necessary sections (Introduction, Geotectonic setting, Materials and Methods, Results, Discussion, Conclusions) have been considered. Moreover, the “Geotectonic setting”, “Materials and Methods” and “Results” sections are divided into sub-sections, providing a further analysis. In addition, all Figures, Diagrams and Tables are consistent with the description provided in the manuscript. However, some changes should be implemented, which will improve the paper. In particular:

Line 29: Please, modify “figure 1” to “Figure 1”. Similarly, apply throughout the manuscript, please.

Lines 58-64: I suggest rearranging this paragraph. In particular, it would be more proper to describe the steps/procedures followed in the research, and then the aim, achieved by the implementation of these steps/procedures. Please, apply.

Line 69: Please, add the corresponding references at the end of the sentence “…during the Late Cretaceous”.

Lines 76-77: Between these lines, I suggest adding a brief paragraph, about the seismicity of the Ionian region, which is directly associated with the Kephalonia right-lateral strike-slip fault, mentioned at the end of Line 76. Please, apply and include the relative references. 

Line 87: “Monopolis et al.” is not consistent with the reference No 33. Moreover, I suggest citing additional and updated references, which justify the significant clockwise rotation of the Ionian region. Please, apply.

Lines 268-269: The Figure 5 caption is briefly described. Please, provide a more detailed description.

Line 380: The “Conclusions” section should be modified. In the current form, it resembles an abstract, while the concluding remarks are vague. The concluding remarks should be solid and comprehensive. Maybe, numbering could be performed. Please, apply.

Author Response

Dear reviewr

Reviewer 2 Report

Good and clearly written MS, which will be of interest to applied geologists and petroleum geologists in the region.

i have  some minor comments for the MS, mostly corrections to figures (labelling on the chromatograms, meaning of depth scale, adding symbols to plots, curiosity question about HF effect on sterane isomerisation prediction, use of R calc when I think the authors mean VRE), and one or two areas of text that I feel should be improved - reason for use of multiple thermal maturity parameters (especially if one is a calibrant and the other the predicted test), and development of thermal maturity parameters.

These minor issues and expect the authors can correct these things without further peer-review.

Author Response

Dear reviewer 2, please see the attachment.

Thank you in advance!

Best regards,

The authors
